# SARI: Simplistic Average and Robust Identification based Noisy Partial Label Learning

## Abstract

Partial label learning (PLL) is a weakly-supervised learning paradigm where each training instance is paired with a set of candidate labels (partial label), one of which is the true label. Noisy PLL (NPLL) relaxes this constraint by allowing some partial labels to not contain the true label, enhancing the practicality of the problem. Our work centers on NPLL and presents a minimalistic framework called SARI that initially assigns pseudo-labels to images by exploiting the noisy partial labels through a weighted nearest neighbour algorithm. These pseudo-label and image pairs are then used to train a deep neural network classifier with label smoothing and standard regularization techniques. The classifier's features and predictions are subsequently employed to refine and enhance the accuracy of pseudo-labels. SARI combines the strengths of Average Based Strategies (in pseudo labelling) and Identification Based Strategies (in classifier training) from the literature. We perform thorough experiments on four datasets and compare SARI against nine NPLL and PLL methods from the prior art. SARI achieves state-of-the-art results in all studied settings, obtaining substantial gains in fine-grained classification and extreme noise settings.

## 1 Introduction

The effectiveness of contemporary deep learning methods heavily relies on the presence of high-quality annotated data, which is costly, arduous, labor-intensive, and time-consuming to gather. Weakly supervised learning is a potential solution to this challenge, which performs training only using partially labeled or noisily labeled examples. It has been widely studied in different forms, including multi-label learning (Zhang & Zhou, 2013), semi-supervised learning (Sohn et al., 2020), noisy label learning (Li et al., 2020) etc. Our paper focuses on a weakly supervised setting called Partial Label Learning (PLL), where each training instance is paired with a set of candidate labels (partial label), out of which one is the true label. PLL commonly occurs in datasets where crowd-sourcing is used for annotations, and the annotators are allowed to select several possible labels if they are uncertain about an instance. It naturally arises in several applications such as web-mining (Huiskes & Lew, 2008; Luo & Orabona, 2010), ecological informatics (Briggs et al., 2012) and multi-modal learning (Chen et al., 2017).

PLL algorithms broadly fall into two categories: Average-based strategies (ABS) and Identification-based strategies (IBS). Average-based strategies assume that every candidate label is equally likely to be the ground truth (Hüllermeier & Beringer, 2005; Cour et al., 2009). Identification-based strategies treat the ground truth as a latent variable. They focus on identifying it using certain criteria and maximizing their estimated probability prediction from the model (Jin & Ghahramani, 2002). Recently, IBS based Deep learning methods have leveraged techniques like contrastive learning (Wang et al., 2022a), consistency regularization (Wu et al., 2022) and prototype learning (Wang et al., 2022a; Xia et al., 2023) to achieve impressive performance. Nevertheless, the fundamental limitation of these methods is that they assume that the correct label is always a part of the partial label.

To overcome this limitation and further the practicality, Noisy PLL (NPLL) was proposed. NPLL allows some partial labels to not contain the correct label. In literature, the problem is also referred to as Unreliable Partial Label Learning (UPLL) (Qiao et al., 2023). A variety of methods have been proposed (Wang et al., 2022b; Qiao et al., 2023; Shi et al., 2023; Lian et al., 2023; Xu et al., 2023)

for NPLL. These efforts are predominantly pursued in an identification-centric approach and are encumbered by the following challenges:

*Complexity:* Majority of the methods (Wang et al., 2022b; Qiao et al., 2023; Xu et al., 2023) augment an existing PLL algorithm for the NPLL task. As the base algorithm may encompass multiple trainable modules, this contributes to increased complexity. For example, Wang et al. (2022b) add three new loss terms over PiCO (Wang et al., 2022a), leading to a total of five loss terms. The number of hyper-parameters significantly increases (e.g. loss weighting terms), making implementation difficult in novel scenarios (e.g., different datasets).

*Need for warm-up :* Most methods (Qiao et al., 2023; Lian et al., 2023; Xu et al., 2023) perform warm-up training using their selected base approach before applying their noise handling technique. Too long or short a warm-up period can negatively impact the performance. Deciding the duration of warm-up training adds another dimension of complexity. Some methods also need a clean validation set to decide the warm-up duration (Shi et al., 2023; Lian et al., 2023).

*Error propagation:* To tackle noisy data, current methods apply a detection cum mitigation technique (Wang et al., 2022b; Lian et al., 2023; Shi et al., 2023). Detection errors are prone to occur and can cause error propagation (Xu et al., 2023).

In this work, we propose a novel framework called SARI, which benefits from the strengths of both the average-based and identification-based methodologies. Broadly, SARI consists of three modules that are applied sequentially in every iteration: Pseudo-labelling, Noise robust learning, and Partial label augmentation. The pioneering work of Hüllermeier & Beringer (2005) establishes that methods with a strong inductive bias, like K-Nearest Neighbours (KNN), can exploit information in the partial labels for better disambiguation. Inspired by this, we assign a pseudo-label for each image using a weighted KNN algorithm. Then, we follow the principle of IBS and train a noise-robust classifier using reliable image and pseudo-label pairs. To impart robustness towards the noisy partial labels, this stage benefits from additional time-tested techniques like label smoothing (Szegedy et al., 2016), consistency regularization (Bachman et al., 2014; Sajjadi et al., 2016) and Mix-up (Zhang et al., 2017). Finally, we apply partial label augmentation. In noisy settings, the partial label may not always contain the ground truth; hence, we allow partial label augmentation based on the confident top-1 predictions of the current classifier.

In a nutshell, our method only has a single trainable module, the encoder+classifier (e.g. a ResNet-18 (He et al., 2016) in our experiments). Additionally, we use standard regularization techniques and a KNN algorithm on the features obtained from the same classifier backbone. We perform comprehensive experiments on four datasets in widely different settings (varying noise rates and number of candidate labels). Albeit simplistic, the proposed framework outperforms existing State-Of-The-Art (SOTA) methods by a significant margin. In contrast to previous methods, SARI manages to preserve a substantial portion of its performance even as the noise ratio increases. Moreover, the performance improvements, compared to other methods, become more pronounced with an increase in both the number of classes and the proportion of noise in partial labels. In fine-grained classification scenarios, SARI yields significant improvements (outperforming all other methods by over 7pp). In summary, our work makes the following contributions:

- We propose a novel framework called SARI, which combines the benefits of both average based and identification based strategies. We do not assert any novelty in terms of architecture; rather, the novelty lies in demonstrating the potential of a simpler alternative for NPLL.

- We present thorough ablation studies and quantitative experiments to support our design choices and demonstrate the efficacy of our approach. We show notable gains over nine SOTA PLL and NPLL methods.

## 2 RELATED WORK

### 2.1 TRADITIONAL PARTIAL LABEL LEARNING

*Identification-based strategies* treat the ground truth as a latent variable and progressively refine the confidence of each candidate label during training. Jin & Ghahramani (2002) use a maximum

likelihood criterion to learn the parameters of a classifier that maximizes the latent label distribution, estimated from the classifier predictions in an EM fashion. Yu & Zhang (2016) propose a maximum margin formulation for PLL, which maximizes the margin between the ground-truth label and all other labels.

*Average-based strategies* treat all candidate labels equally while distinguishing between the candidate and non-candidate labels. Early work by Hüllermeier & Beringer (2005) extends the K-Nearest neighbours (KNN) algorithm to tackle PLL by employing majority voting among the candidate labels of neighbours. Zhang & Yu (2015) also utilize KNN to classify any unseen instance based on minimum error reconstruction from its nearest neighbours. Cour et al. (2009) maximize the average output of candidate labels and minimize the output of non-candidate labels in parametric models. SARI takes a cue from the KNN-based ABS approach from the 2000s (Hüllermeier & Beringer, 2005) and uses a variation for the pseudo-labelling step.

## 2.2 Deep Partial Label Learning

With the advent of Deep Learning, a variety of identification-based strategies that employ a neural network backbone have been proposed. Yao et al. (2020) temporally ensemble the predictions at different epochs to disambiguate the partial label. Lv et al. (2020) use self-training to update the model and progressively identify the true label. Feng et al. (2020) formulate the generation process of partial labels and develop two classifiers: one risk-consistent and the other another classifier-consistent. Wen et al. (2021) propose leverage weighted loss, a family of loss functions that generalize the loss functions proposed by earlier works (Jin & Ghahramani, 2002; Cour et al., 2009; Lv et al., 2020). Wang et al. (2022a) present PiCO, a framework that learns discriminative representations by leveraging contrastive learning and prototypical classification. Wu et al. (2022) present a simple idea that uses consistency regularization on the candidate set and supervised learning on the non-candidate set. Xia et al. (2023) upgrade PiCO and introduce self-training and prototypical alignment to achieve noteworthy results. However, none of these methods account for the potential of noise within partial labels, which is the primary focus of our work.

## 2.3 Noisy Partial Label Learning

Earlier works assume that the correct label is always a part of the partial label, which limits the practicality of the problem. Hence, some recent works have diverted attention to NPLL that relaxes this condition, and allows some partial labels not to contain the correct label. Qiao et al. (2023) perform disambiguation to move incorrect labels from candidate labels to non-candidate labels but also refinement to move correct labels from non-candidate labels to candidate labels. Shi et al. (2023) separate the dataset into reliable and unreliable sets and then perform label disambiguation-based training for the former and semi-supervised learning for the latter. Lian et al. (2023) iteratively detect and purify the noisy partial labels, thereby reducing the noise level in the dataset. Wang et al. (2022b) extend PiCO to tackle noisy PLL by performing distance-based clean sample selection and learning a robust classifier by semi-supervised contrastive learning. Xu et al. (2023) reduce the negative impact of detection errors by carefully aggregating the partial labels and model outputs.

Unlike the aforementioned identification-focused approaches, Lv et al. (2023) propose Average Partial Loss (APL), a family of loss functions that achieve promising results for both PLL and noisy PLL. Moreover, they provide insights into how ABS can enhance IBS when used for warm-up training. Our work builds upon this intuition by alternating between ABS and IBS. Our findings demonstrate that opting for conventional choices, such as employing K-nearest neighbours (KNN) for ABS and utilizing cross-entropy loss with standard regularization techniques for IBS, yields state-of-the-art performance.

## 3 Methodology

Figure 1 depicts the three modules of SARI that are applied sequentially in every iteration: Pseudo Labelling (3.1.1), Noise robust learning (3.1.2), Partial label augmentation (3.1.3). First, we assign pseudo-labels to all images using a weighted KNN. Now, for each unique class, we select the $m$ most reliable image-label pairs. We use these reliable image-label pairs for training the classifier. To be resilient towards the potential noise in the pseudo-labelling stage, we leverage label smoothing,

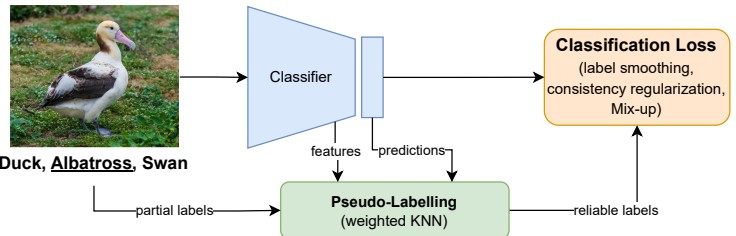

Figure 1: Illustration of SARI. Pseudo-labelling is performed using a weighted KNN to obtain reliable image-label pairs. These pairs are used to train a classifier using label smoothing, consistency regularization, and Mix-up. Finally, the confident top-1 model predictions are used to augment the partial label for the upcoming iteration.

consistency regularization and Mix-up during training. Finally, we optionally augment the partial labels with the confident top-1 predictions of the classifier. The features from the updated classifier are then used for the next stage of pseudo-labelling. We hypothesize that, as training progresses, more number of samples will be assigned the correct label as the pseudo-label, which will help learn an improved classifier. Pseudo-code of SARI is available in Algorithm 1. We now provide a detailed description of each of the mentioned steps.

## 3.1 PROBLEM SETUP

The problem of partial label learning (PLL) is defined using the following setup. Let $\mathcal{X}$ be the input space, and $\mathcal{Y} = \{1, 2, \ldots, C\}$ be the output label space. We consider a training dataset $\mathcal{D} = \{(\boldsymbol{x}_i, Y_i)\}_{i=1}^n$, where each tuple comprises of an image $\boldsymbol{x}_i \in \mathcal{X}$ and a partial label $Y_i \subset \mathcal{Y}$. Similar to a supervised learning setup, PLL aims to learn a classifier that predicts the correct output label. However, the training dataset in PLL contains high ambiguity in the label space, which makes the training more difficult when compared to supervised learning. A basic assumption of PLL is that the correct label $y_i$ is always present in the partial label, i.e., $\forall i \; y_i \in Y_i$. In noisy PLL, we relax this constraint, and the correct label potentially lies outside the candidate set, i.e., $\exists i \; y_i \notin Y_i$. This practical assumption adds another layer of ambiguity.

### 3.1.1 PSEUDO-LABELLING

For each sample $(\boldsymbol{x}_i, Y_i)$, we pass the image $\boldsymbol{x}_i$ through the encoder $f(\cdot)$ to obtain the feature representation $\boldsymbol{z}_i$. Then, we select its $K$ nearest neighbours from the entire dataset. Let $\mathcal{N}_i$ denote the set of $K$ nearest neighbours of $\boldsymbol{x}_i$. We then compute a pseudo-label $\hat{y}_i$ using weighted KNN as follows.

$$\hat{y}_i = \arg\max_c \sum_{\boldsymbol{x}_k \in \mathcal{N}_i} \mathbb{I}[c \in Y_k] \cdot d_{ik} \qquad c \in [C], \tag{1}$$

where $d_{ik}$ is the cosine distance between $\boldsymbol{z}_i$ and $\boldsymbol{z}_k$ ($\boldsymbol{x}_k \in \mathcal{N}_i$). We assume that these pseudo-labels are less ambiguous than the partial labels supposing samples in the neighbourhood have the same class label. Now, we approximate the class posterior probabilities $\hat{q}_c$ via KNN, inspired by the Noisy Label Learning literature (Ortego et al., 2021).

$$\hat{q}_c(\boldsymbol{x}_i) = \frac{1}{K} \sum_{\boldsymbol{x}_k \in \mathcal{N}_i} \mathbb{I}[\hat{y}_k = c] \cdot d_{ik} \qquad c \in [C] \tag{2}$$

Using these posterior probabilities, we select a maximum of $m$ most reliable image-label pairs for each class. We do this to ensure a near uniform distribution of samples across all classes. Using $\hat{q}_c(\boldsymbol{x}_i)$, we first compute $a_1, a_2, \ldots, a_C$ as follows.

$$a_c = \sum_{i=1}^n \mathbb{I}\left[c = \arg\max_{c'} \hat{q}_{c'}(\boldsymbol{x}_i) \; \& \; c \in Y_i\right], \qquad c \in [C] \tag{3}$$

---

**Algorithm 1** Pseudo-code of SARI

---

**Input:** Dataset $\mathcal{D}$, encoder $f(\cdot)$, classifier $h(\cdot)$, epochs $T_{max}$, mini-batch size $B$, hyper-parameters $k, \delta, \zeta, r, \lambda_{max}, \lambda_{min}$
  1: **for** $t = 1, 2, \ldots, T_{max}$ **do**
  2:      Compute psuedo-labels $\hat{y}_i$ for all samples $\boldsymbol{x}_i$ in the dataset using Eq.(1);
  3:      Compute the posterior probability vectors using Eq.(2);
  4:      Compute the maximum samples to be selected per class $m$ using Eq.(3);
  5:      Build the Reliable image-label pairs set $\mathcal{T}$ using Eq.(4);
  6:      Shuffle $\mathcal{T}$ into $\frac{\mathcal{T}}{B}$ mini-batches;
  7:      **for** $b = 1, 2, \ldots, \frac{\mathcal{T}}{B}$ **do**
  8:          Update parameters of $f(\cdot)$ and $h(\cdot)$ by minimizing loss in Eq.(10);
  9:      **end for**
10:      Perform Partial label augmentation using Eq.(11);
11: **end for**
**Output:** parameters of $f(\cdot)$ and $h(\cdot)$

---

Thus, $a_c$ is the number of samples for which $c$ is the highest probable class as per posterior probability and $c$ is also in the partial label of those samples. We choose $m$ as the $\delta$-quantile of $[a_1, a_2, \ldots, a_C]$. For instance, when $\delta = 0$, $m = \min([a_1, a_2, \ldots, a_C])$.

As training progresses and the classifier learns, the per-class agreements $a_c$ also increase — therefore $m$ increases and, consequently, the number of reliable pseudo-labels. Once $m$ is finalized, we need to select $m$ most reliable images for each class. For class $c$, the images for which the cross entropy loss $\mathcal{L}_{ce}$ falls below threshold $\gamma_c$ are chosen as reliable images. Thus, the set of reliable pairs belonging to the $c$-th class (denoted as $\mathcal{T}_c$) is found as follows.

$$\mathcal{T}_c = \{(\boldsymbol{x}_i, c) \mid \mathcal{L}_{ce}(\hat{\boldsymbol{q}}(\boldsymbol{x}_i), c) < \gamma_c, c \in Y_i, i \in [n]\} \qquad c \in [C] \tag{4}$$

For the $c$-th class, the threshold $\gamma_c$ is dynamically defined such that a maximum of $m$ samples can be selected per class. Note that it is possible for a sample $\boldsymbol{x}_i$ to belong to different $\mathcal{T}_c$. In that case, we choose the one that has the highest $\hat{q}_c(\boldsymbol{x}_i)$. Finally, we form the reliable image-label pair set $\mathcal{T}$ as

$$\mathcal{T} = \bigcup_{c=1}^{C} \mathcal{T}_c = \{(\boldsymbol{x}_i, \tilde{y}_i)\}_{i=1}^{\tilde{n}} \tag{5}$$

where $\tilde{y}_i$ is the reliable pseudo-label for $\boldsymbol{x}_i$ and $\tilde{n}$ is the number of selected reliable samples.

### 3.1.2 Noise Robust Learning

We train a noise-robust classifier $h(\cdot)$ by leveraging label smoothing (Szegedy et al., 2016), consistency regularization (Bachman et al., 2014) and mix-up. Label smoothing uses a positively weighted average of the hard training labels and uniformly distributed soft labels to generate the smoothed labels. Let $\mathbf{e}_{\tilde{y}_i}$ be $C$-dimensional one-hot encoding of label $\tilde{y}_i$ such that its $\tilde{y}_i^{th}$ element is one and rest of the entries are zero. The corresponding smoothed label $\mathbf{e}_{\tilde{y}_i}^{LS,r}$ is:

$$\mathbf{e}_{\tilde{y}_i}^{LS,r} = (1 - r) \cdot \mathbf{e}_{\tilde{y}_i} + \frac{r}{C} \cdot \mathbf{1} \tag{6}$$

Then the per-sample classification loss is formulated as:

$$\mathcal{L}_{ce}(\boldsymbol{x}_i, \mathbf{e}_{\tilde{y}_i}^{LS,r}) = -(1-r)\log(h^{\tilde{y}_i}(f(\boldsymbol{x}_i))) - \frac{r}{C}\sum_{j=1}^{C}\log(h^j(f(\boldsymbol{x}_i))) \tag{7}$$

Although label smoothing introduces an additional hyper-parameter $r$, fixing it to a high value that is greater than the true noise rate ($r = 0.5$ in the experiments) improves performance (Lukasik et al., 2020). We also use mix-up (Zhang et al., 2017) for further noise tolerance. Consider a pair of samples $(\boldsymbol{x}_i, \mathbf{e}_{\tilde{y}_i}^{LS,r})$ and $(\boldsymbol{x}_j, \mathbf{e}_{\tilde{y}_j}^{LS,r})$. We create the mix-up image by linearly interpolating with a factor $\alpha_{mix} \sim Beta(\zeta, \zeta)$ where $\zeta$ is a parameter in the beta distribution and define the mix-up loss below.

$$mix_x(\boldsymbol{x}_i, \boldsymbol{x}_j) = \alpha_{mix}\boldsymbol{x}_i + (1 - \alpha_{mix})\boldsymbol{x}_j \tag{8}$$

$$\mathcal{L}_{mix}(\boldsymbol{x}_i, \boldsymbol{x}_j) = \alpha_{mix}\mathcal{L}_{ce}(mix_x(\boldsymbol{x}_i, \boldsymbol{x}_j), \mathbf{e}_{\tilde{y}_i}^{LS,r}) + (1 - \alpha_{mix})\mathcal{L}_{ce}(mix_x(\boldsymbol{x}_i, \boldsymbol{x}_j), \mathbf{e}_{\tilde{y}_j}^{LS,r}) \quad (9)$$

Finally, we include consistency regularization in the complete training objective. Consistency regularization is a simple idea that the model should output similar predictions when fed with perturbed versions of the same image. We implement a variation by assigning the same pseudo-label to both weak and strong augmentations of an image.

$$\mathcal{L}_{final} = \sum_{i,j,j'} \mathcal{L}_{mix}(aug_w(\boldsymbol{x}_i), aug_w(\boldsymbol{x}_j)) + \mathcal{L}_{mix}(aug_s(\boldsymbol{x}_i), aug_s(\boldsymbol{x}_{j'})) \qquad i, j, j' \in [\tilde{n}] \quad (10)$$

where $aug_w(\cdot)$ and $aug_s(\cdot)$ represent weak and strong augmentation function respectively.

### 3.1.3 PARTIAL LABEL AUGMENTATION

In Pseudo-Labelling (Eq.4), for every sample image, we select its reliable pseudo-label only from the partial label. This limits the number of correct samples that can be selected in the noisy PLL case. To overcome this issue, we include the highest probability class of the current model prediction in the partial label for the next iteration if it is greater than a threshold $\lambda^t$. $\lambda^t$ is a decaying threshold and can be interpreted as balancing between precision and recall. Initially, the high threshold implies that few accurate predictions are used to avoid error propagation. The threshold is later decreased to allow more samples to be de-noised.

$$Y_i^{t+1} = \begin{cases} Y_i \cup \{\arg\max_c \ h_t^c(f_t(aug_w(\boldsymbol{x}_i)))\}, & \text{if } \max_c h_t^c(f_t(aug_w(\boldsymbol{x}_i))) > \lambda^t \\ Y_i, & \text{otherwise.} \end{cases} \quad (11)$$

Here, $f_t$ and $h_t$ are the encoder and classifier at the $t^{th}$ iteration. It is important to note here that at iteration $t + 1$, we include the highest probability class in the true partial label set $Y_i$ to get $Y_i^{t+1}$. Unlike previous works (Lian et al., 2023; Xu et al., 2023), we do not directly disambiguate the partial label. Instead, we merely include a possible candidate and let the first stage select the label. This prevents error propagation to subsequent iterations.

## 4 EXPERIMENTS

### 4.1 EXPERIMENTAL SETUP

**Datasets** We construct the PLL and NPLL benchmark for four different datasets: CIFAR-10, CIFAR-100, CIFAR-100H (Krizhevsky et al., 2009) and CUB-200 (Welinder et al., 2010). We follow the same dataset generation process of Wang et al. (2022b). The dataset generation process is governed by two parameters: the partial rate ($q$) and the noise rate ($\eta$). The partial rate $q$ represents the probability of an incorrect label to be present in the candidate set. The probability of excluding the correct label from the candidate set is referred to by the noise rate $\eta$. For CIFAR10, we experiment with $q \in \{0.1, 0.3, 0.5\}$, and for other datasets (with a larger number of classes) we select $q$ from the set $\{0.01, 0.03, 0.05\}$. We conduct experiments using different values of $\eta \in \{0.1, 0.2, 0.3, 0.4, 0.5\}$, where values of $\eta$ exceeding 0.3 indicate high noise scenarios.

**Baselines** We compare SARI against four PLL and five NPLL approaches, a total of nine state-of-the-art methods. The PLL methods include 1) RC (Feng et al., 2020): a risk-consistent classifier for PLL; 2) LWS (Wen et al., 2021): trades of loss between candidate labels and non-candidate labels; 3) PICO (Wang et al., 2022a): learns discriminative representations using contrastive and prototype learning 4)CRDPLL (Wu et al., 2022): adapts consistency regularization for PLL. The NPLL methods include 1) PICO+ (Wang et al., 2022b): extends PiCO for NPLL by including additional losses; 2) FREDIS (Qiao et al., 2023): performs partial label disambiguation and refinement 3) IRNet (Lian et al., 2023): detects and corrects the partial labels iteratively; 4) UPLLRS (Shi et al., 2023): proposes a two-stage framework for dataset separation and disambiguation; 3) ALIM (Xu et al., 2023): aggregates the partial labels and model outputs using two variants of normalization (Scale and OneHot). If available, we directly reference the numbers from the existing literature, and whenever possible, we present results obtained by customizing the publicly available official codebase to suit the specific configuration.

**Implementation Details.** To ensure a fair comparison, we use ResNet-18 (He et al., 2016) as the encoder $f(\cdot)$ and $h(\cdot)$ as the linear classifier. Other methods either use ResNet-18 (He et al.,

Table 1: Comparison of SARI with the previous state-of-the-art methods across various partial and noise rates. The best highest accuracy in each setting is highlighted in bold.

| CIFAR-10 | $q = 0.1$ | | | $q = 0.3$ | | | $q = 0.5$ | | |
|---|---|---|---|---|---|---|---|---|---|
| | $\eta = 0.1$ | $\eta = 0.2$ | $\eta = 0.3$ | $\eta = 0.1$ | $\eta = 0.2$ | $\eta = 0.3$ | $\eta = 0.1$ | $\eta = 0.2$ | $\eta = 0.3$ |
| RC | 80.87±0.30 | 78.22±0.23 | 75.24±0.17 | 79.69±0.37 | 75.69±0.63 | 71.01±0.54 | 72.46±1.51 | 59.72±0.42 | 49.74±0.70 |
| LWS | 82.97±0.24 | 79.46±0.09 | 74.28±0.79 | 80.93±0.28 | 76.07±0.38 | 69.70±0.72 | 70.41±2.68 | 58.26±0.28 | 39.42±3.09 |
| FREDIS | 90.57±0.23 | - | 84.35±0.20 | 89.02±0.15 | - | 81.02±0.60 | 87.42±0.21 | - | 65.15±0.13 |
| PiCO | 90.78±0.24 | 87.27±0.11 | 84.96±0.12 | 89.71±0.18 | 85.78±0.23 | 82.25±0.32 | 88.11±0.29 | 82.41±0.30 | 68.75±2.62 |
| CRDPLL | 93.48±0.17 | 89.13±0.39 | 86.19±0.48 | 92.73±0.19 | 86.96±0.21 | 83.40±0.14 | 91.10±0.07 | 82.30±0.46 | 73.78±0.55 |
| IRNet | 93.44±0.21 | 92.57±0.25 | 92.38±0.21 | 92.81±0.19 | 92.18±0.18 | 91.35±0.08 | 91.51±0.05 | 90.76±0.10 | 86.19±0.41 |
| PiCO+ | 94.48±0.02 | 94.74±0.13 | 94.43±0.19 | 94.02±0.03 | 94.03±0.01 | 92.94±0.24 | 93.56±0.08 | 92.65±0.26 | 88.21±0.37 |
| UPLLRS | 95.16±0.10 | - | 94.65±0.23 | 94.32±0.21 | - | 93.85±0.31 | 92.47±0.19 | - | 91.55±0.38 |
| ALIM-Scale | 95.71±0.01 | 95.50±0.08 | 95.35±0.13 | 95.31±0.16 | 94.77±0.07 | 94.36±0.03 | 94.71±0.04 | 93.82±0.13 | 90.63±0.10 |
| ALIM-Onehot | 95.83±0.13 | 95.86±0.15 | 95.75±0.19 | 95.52±0.15 | 95.41±0.13 | 94.67±0.21 | 95.19±0.24 | 93.89±0.21 | 92.26±0.29 |
| SARI | **96.28±0.05** | **96.17±0.18** | **95.90±0.20** | **95.96±0.08** | **95.76±0.12** | **95.43±0.17** | **95.52±0.13** | **95.89±0.14** | **94.18±0.10** |

| CIFAR-100 | $q = 0.01$ | | | $q = 0.03$ | | | $q = 0.05$ | | |
|---|---|---|---|---|---|---|---|---|---|
| | $\eta = 0.1$ | $\eta = 0.2$ | $\eta = 0.3$ | $\eta = 0.1$ | $\eta = 0.2$ | $\eta = 0.3$ | $\eta = 0.1$ | $\eta = 0.2$ | $\eta = 0.3$ |
| RC | 52.73±1.05 | 48.59±1.04 | 45.77±0.31 | 52.15±0.19 | 48.25±0.38 | 43.92±0.37 | 46.62±0.34 | 45.46±0.21 | 40.31±0.55 |
| LWS | 56.05±0.20 | 50.66±0.59 | 45.71±0.45 | 53.59±0.45 | 48.28±0.44 | 42.20±0.49 | 45.46±0.44 | 39.63±0.80 | 33.60±0.64 |
| FREDIS | 64.73±0.28 | - | 59.42±0.18 | - | - | - | 66.43±0.22 | - | 56.15±0.18 |
| PiCO | 68.27±0.08 | 62.24±0.31 | 58.97±0.09 | 67.38±0.09 | 62.01±0.33 | 58.64±0.28 | 67.52±0.43 | 61.52±0.28 | 58.18±0.65 |
| CRDPLL | 68.12±0.13 | 65.32±0.34 | 62.94±0.28 | 67.53±0.07 | 64.29±0.27 | 61.79±0.11 | 67.17±0.04 | 64.11±0.42 | 61.03±0.43 |
| IRNet | 71.17±0.14 | 70.10±0.28 | 68.77±0.28 | 71.01±0.43 | 70.15±0.17 | 68.18±0.30 | 70.73±0.09 | 69.33±0.51 | 68.09±0.12 |
| PiCO+ | 75.04±0.18 | 74.31±0.02 | 71.79±0.17 | 74.68±0.19 | 73.65±0.23 | 69.97±0.01 | 73.06±0.16 | 71.37±0.16 | 67.56±0.17 |
| UPLLRS | 75.73±0.41 | - | 71.72±0.39 | - | - | - | 74.73±0.24 | - | 70.31±0.22 |
| ALIM-Scale | 77.37±0.32 | 76.81±0.05 | 76.45±0.30 | 77.60±0.18 | 76.63±0.19 | 75.92±0.14 | 76.86±0.23 | 76.44±0.12 | 75.67±0.17 |
| ALIM-Onehot | 76.52±0.19 | 76.55±0.24 | 76.09±0.23 | 77.27±0.23 | 76.29±0.41 | 75.29±0.57 | 76.87±0.20 | 75.23±0.42 | 74.49±0.61 |
| SARI | **80.90±0.50** | **80.45±0.49** | **79.78±0.13** | **80.33±0.04** | **79.40±0.50** | **78.52±0.24** | **80.00±0.25** | **79.08±0.22** | **77.87±0.29** |

Table 2: PLL Experiments

| CIFAR-100 | $q = 0.01$ | $q = 0.05$ | $q = 0.1$ |
|---|---|---|---|
| RC | 75.36 ± 0.06 | 74.44 ± 0.31 | 73.79 ± 0.29 |
| LWS | 64.55 ± 1.98 | 50.19 ± 0.34 | 44.93 ± 1.09 |
| PiCO | 73.78 ± 0.15 | 72.78 ± 0.38 | 71.55 ± 0.31 |
| CRDPLL | 79.74 ± 0.07 | 78.97 ± 0.13 | 78.51 ± 0.24 |
| PiCO+ | 76.29 ± 0.42 | 76.17 ± 0.18 | 75.55 ± 0.21 |
| SARI | **81.46 ± 0.13** | **81.00 ± 0.26** | **80.77 ± 0.12** |

Table 3: Extreme noise experiments

| CIFAR-100 | $q = 0.05, \eta = 0.4$ | $q = 0.05, \eta = 0.5$ |
|---|---|---|
| FREDIS | - | 48.05 ± 0.20 |
| PiCO | 44.17 ± 0.08 | 35.51 ± 1.14 |
| CRDPLL | 57.10 ± 0.24 | 52.10 ± 0.36 |
| UPLLRS | - | 64.78 ± 0.53 |
| PiCO+ | 66.41 ± 0.58 | 60.50 ± 0.99 |
| ALIM-Scale | 74.98 ± 0.16 | 72.26 ± 0.25 |
| ALIM-Onehot | 71.76 ± 0.56 | 69.59 ± 0.62 |
| SARI | **76.72 ± 0.41** | **74.79 ± 0.40** |

2016) or a stronger backbone. More specifically, FREDIS uses ResNet32 and UPLLRS uses a WideResNet28×2 (Zagoruyko & Komodakis, 2016). Strong augmentation function utilizes autoaugment (Cubuk et al., 2018) and cutout (DeVries & Taylor, 2017). Faiss (Johnson et al., 2019) is used to speed up nearest neighbour computation. Irrespective of the partial and noise rate, we set $K = 15, \delta = 0.25, \zeta = 1.0, r = 0.5$, and $\lambda_t$ linearly decreases in $[0.45, 0.35]$. For CIFAR datasets, we choose the SGD optimizer with a momentum of 0.9 and set the weight decay to 0.001. The initial learning rate is set to 0.1 and decayed using a cosine scheduler, and the model is trained for 500 epochs. For CUB-200, as per the existing works, we initialize the encoder with ImageNet-1K (Russakovsky et al., 2015) pre-trained weights. We choose the SGD optimizer with a momentum of 0.9 and set the weight decay to 0.0005. We set the initial learning rate to 0.05 and decay it by 0.2 using the step scheduler at epochs $[60, 120, 160, 200]$ and train for 250 epochs. To eliminate the randomness of the results, we ran each experiment three times and report the average result and standard deviation on the test set. All experiments are implemented with PyTorch and carried out with NVIDIA GeForce RTX 2080 Ti GPU.

## 4.2 RESULTS AND DISCUSSION

**SOTA comparisons:** Table 1 compares SARI with the SOTA methods on CIFAR-10 and CIFAR-100 datasets. The results are presented with three different partial-ratio and noise-ratio values, comprising a total of nine different combinations. SARI surpasses all other methods in all settings, providing clear evidence of its effectiveness. It is also apparent that conventional PLL approaches struggle to generalize effectively in noisy settings, highlighting the necessity for noise-resistant adaptations.

Table 4: Fine-grained datasets

| Dataset | CIFAR-100H | CUB-200 | CUB-200 | CUB-200 | CUB-200 |
|---|---|---|---|---|---|
| $q$ | 0.5 | 0.05 | 0.01 | 0.05 | 0.1 |
| $\eta$ | 0.2 | 0.2 | 0.0 | 0.0 | 0.0 |
| PiCO | $59.8 \pm 0.25$ | $53.05 \pm 2.03$ | $74.14 \pm 0.24$ | $72.17 \pm 0.72$ | $62.02 \pm 1.16$ |
| PiCO+ | $68.31 \pm 0.47$ | $60.65 \pm 0.79$ | - | - | - |
| ALIM-Scale | $73.42 \pm 0.18$ | $68.38 \pm 0.47$ | - | - | - |
| ALIM-Onehot | $72.36 \pm 0.20$ | $63.91 \pm 0.35$ | - | - | - |
| SARI (Ours) | $\mathbf{75.16 \pm 0.59}$ | $\mathbf{75.76 \pm 0.27}$ | $\mathbf{80.71 \pm 0.21}$ | $\mathbf{79.46 \pm 0.25}$ | $\mathbf{78.73 \pm 0.36}$ |

Table 5: Ablation studies

| Mix-up | CR | $q = 0.05, \eta = 0.3$ | $q = 0.05, \eta = 0.5$ |
|---|---|---|---|
| False | False | 73.3 | 71.41 |
| False | True | 74.33 | 72.51 |
| True | False | 75.91 | 73.12 |
| True | True | 77.61 | 75.15 |

(a) Impact of Mix-up and Consistency Regularization

| $K$ | $q = 0.05, \eta = 0.3$ | $q = 0.05, \eta = 0.5$ |
|---|---|---|
| 10 | 77.87 | 74.51 |
| 15 | 77.96 | 75.15 |
| 25 | 78.29 | 74.98 |
| 50 | 77.01 | 75.60 |

(b) Impact of K

| $\delta$ | 0.0 | 0.25 | 0.50 | 0.75 |
|---|---|---|---|---|
| $q = 0.05, \eta = 0.3$ | 75.92 | 77.61 | 78.22 | 77.62 |
| $q = 0.05, \eta = 0.5$ | 73.50 | 75.15 | 74.72 | 72.24 |

(c) Impact of $\delta$

| $r$ | 0.6 | 0.5 | 0.4 | 0.0 |
|---|---|---|---|---|
| $q = 0.05, \eta = 0.3$ | 76.87 | 77.61 | 78.19 | 75.64 |
| $q = 0.05, \eta = 0.5$ | 74.17 | 75.15 | 74.41 | 65.63 |

(d) Impact of label smoothing rate $r$

The performance of all methods drops with an increase in the size of the candidate label set (higher $q$) and with an increase in noise ratio. A key aspect is to observe the performance across the first and last column of Table 1 (the easiest vs. the most complex setting). In several methods, the performance significantly drops; for instance, on CIFAR-10, the performance of LWS drops from 82.97 to 39.42. SARI and ALIM fare better in this regard and are able to retain model performance with increased complexity. Another noteworthy observation is that the performance improvement of SARI becomes more pronounced in CIFAR-100 compared to the CIFAR-10 dataset.

**PLL results:** While our primary focus is on NPLL, assessing performance in the noise-free setting (PLL) is crucial to ensure that the adaptations for noise robustness do not negatively affect zero noise performance. As most NPLL methods do not provide results in noise-free settings, we limit our comparisons primarily to PLL methods. In Table 2, we evaluate SARI on the CIFAR-100 PLL benchmark. The results indicate that SARI also achieves superior performance in the absence of noise.

**Extreme noise results:** In Table 3, we present results in extremely noisy settings (e.g., $\eta = 0.5$, the correct label is absent from half of the partial labels). For most methods, the performance drastically drops in extreme noise settings. For example, for PICO+ the performance drops from 76.17 ($\eta = 0$) to 60.50 ($\eta = 0.5$). For CRDPLL, the drop is from 78.97($\eta = 0$) to 52.10 ($\eta = 0.5$). ALIM-Onehot also observes a significant drop. ALIM-Scale and SARI fare well in extreme noise settings, with SARI obtaining the best results.

**Fine-grained classification:** We first evaluate SARI on the CIFAR-100H dataset, which considers label hierarchy and divides the CIFAR-100 dataset into 20 coarse labels (5 fine-grained labels per coarse label). When curating the NPLL benchmark, the candidate sets are restricted to lie within the same coarse label. SARI also obtains the best results in this modified setting (Table 4).

We also evaluate the performance of SARI on the CUB-200 bird classification dataset. Since all the images exclusively showcase birds, the candidate labels are intricately linked, presenting a heightened challenge for disambiguation. Table 4 presents the results with different $q$ and $\eta$ values. SARI outperforms other approaches by a significant margin, bringing over 7pp improvement over ALIM-Scale at $q = 0.05$ and $\eta = 0.2$.

### 4.3 ABLATION STUDIES

In Table 5, we perform ablations using two different noise ratios: $\eta = 0.3$ and $\eta = 0.5$), while keeping $q = 0.05$. All the ablations are performed using the CIFAR-100 dataset.

**Influence of regularization components:** In table 5a, we study the impact of using Mix-up and Consistency Regularization (CR). Removing Mix-up causes a decrease of around $2 - 3$pp, and the exclusion of CR causes a decrease of about 1pp, which validates their role in our framework. Another noteworthy observation is that their combined usage generates a more substantial effect.

**Impact of $K$ in KNN:** We vary the value of $k$ from the set $\{10, 15, 25, 50\}$. We find that performance remains similar across different "k" values, with statistically insignificant differences. We use $k = 15$ for all our experiments.

**Impact of $\delta$ parameter:** $\delta$ controls the number of samples chosen per class to construct the reliable set. In table 5c, $\delta = 0.5$ yields the best results for $\eta = 0.3$, while $\delta = 0.25$ produces the best results for $\eta = 0.5$. Lower $\delta$ implies that a smaller but more confident reliable set is constructed, which leads to better performance in a high noise setting.

**Impact of label smoothing:** In Table 5d, $r = 0, 4$ yields the best results for $\eta = 0.3$, while $r = 0.5$ produces the best results for $\eta = 0.5$. Considering that elevated noise levels lead to increased noise in pseudo-labelling, a higher label smoothing rate appears to confer advantages. Moreover, when $r$ is zero, there is a significant drop in the accuracy in the extreme noise setting. The result strongly establishes the noise-robustness imparted by label smoothing in the NPLL context.

## 5 CONCLUSION

We propose SARI, a simplistic framework for NPLL that iteratively performs pseudo-labeling and classifier training. Pseudo-labeling is done using weighted KNN, and a classifier is trained using the labeled samples while leveraging label smoothing, consistency regularization, and Mix-up. SARI achieves SOTA results in a comprehensive set of experiments, varying partial rates and noise rates. The gains are more pronounced with an increase in noise and the number of classes. Notably, SARI excels in fine-grained classification tasks. Unlike many previous methods, SARI preserves its performance as the ambiguity in the dataset increases.

We provide thorough ablation studies to shed light on the functioning of the SARI framework. Our analysis underscores the consistent contributions of both consistency regularization and Mix-up to overall performance improvements. Additionally, we highlight the enormous benefits of label smoothing, particularly in high-noise scenarios. We believe that the SARI framework and the insights presented in this study would be helpful for the research community across a wide range of practical applications.

### REPRODUCIBILITY STATEMENT

We commit to release the code for SARI (our framework) and the NPLL dataset curation upon acceptance. Our dataset curation procedure is the same as that of Wang et al. (2022b).

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
