# OpenReview forum: "SARI: SIMPLISTIC AVERAGE AND ROBUST IDENTIFICATION BASED NOISY PARTIAL LABEL LEARNING"
_ICLR.cc/2024/Conference — ICLR 2024 Conference Withdrawn Submission_

### Official Review · Reviewer_WNqE · 2023-10-25

**Soundness:** 3 good
**Presentation:** 3 good
**Contribution:** 2 fair
**Rating:** 6
**Confidence:** 5

**Summary:**

To deal with the problem of noisy partial label learning (NPLL), this paper proposes a framework called SARI via combining the strengths of Average Based Strategies and Identification Based Strategies. Experimental results validate its effectiveness.

**Strengths:**

1.This paper proposes an effective frameworks, which is illustrated from the experimental results.

2.The figures and tables are well presented.

3.The proposed method is easy to understand and follow.

**Weaknesses:**

1.My main concern is the novelty of this paper. As it claims, the novelty lies in the potential of a simpler alternatives for NPLL instead of the architecture.

2.The simplicity of the proposed framework is partly reflected in no demand for warm-up. However, one of the key techniques is to perform pseudo-labeling via a weighted KNN algorithm, and the KNN algorithm often needs an effective feature extractor to perform well. Without the process like warm-up or pre-training, we could not obtain such an effective feature extractor.

**Questions:**

1.As it is mentioned above, how can the pseudo-labeling be implemented well in the proposed method with the weighted KNN? Please give more details about it.

---

> ### Author Response · Authors · 2023-11-17
>
> We thank the reviewer for their positive feedback and address their concerns below:
>
> * **Q1: Novelty of the proposed framework.** Please refer to the global response. If minimal and existing ideas work well, it would be unjust to push for unnecessary complexities and fancy architectures.
>
> * **Q2: Need for pre-training to obtain effective feature extractor.**
>     1. We also initially thought that an unsupervised pre-training strategy would be required for effective feature extraction. We employed a few epochs of MoCo to learn the initial set of features (giving a warm start). However, we found no difference in performance with and without such unsupervised pre-training. We illustrate the results below.
>     1. Recent studies [G] suggest that a randomly initialised backbone can also provide good features due to its inductive bias. We hypothesize that this is the reason that the network is able to learn in the initial training phase.
>     1. Further, we highlight that, for the other datasets: CUB-200, Treeversity, Benthic and Plankton, we initialize the network with Imagenet pre-trained weights (in accordance with prior methods) which provides good initial features.
>     1. We would also like to clarify that warm-up in other methods is not unsupervised pre-training. For example, ALIM initially trains PiCO for several epochs (calling it warm-up), and then applies their own ideas on top of this pre-trained model. Hence their method at least needs the complexity of PiCO.
>
>     | CIFAR-100| $q=0.05$, $\eta=0.3$ | $q=0.05$, $\eta=0.5$ |
>     | --- | ----------- | -------|
>     | With MoCo |76.76 |74.29 |
>     | Without MoCo|77.61 | 75.15 |
>
> [G] Ulyanov, Dmitry, Andrea Vedaldi, and Victor Lempitsky. Deep image prior. CVPR. 2018.

---

> > ### Comment · Reviewer_WNqE · 2023-11-23
> >
> > Thank you for your response. It is suggested to further explore the motivation to integrate existing tricks and provide more insight into the problem.  Otherwise, the entire article reads with a sense of patchwork. Consequently, I will maintain my current scoring.

---

### Official Review · Reviewer_EnLY · 2023-10-29

**Soundness:** 2 fair
**Presentation:** 3 good
**Contribution:** 2 fair
**Rating:** 3
**Confidence:** 4

**Summary:**

This manuscript considers a new setting of PLL, i.e., NPLL, and presents a minimalistic framework called SARI that initially assigns pseudo-labels to images by exploiting the noisy partial labels through a weighted nearest neighbour algorithm. The authors conduct some experiments to validate the effectiveness of the proposed method.

**Strengths:**

The manuscript is well-organized, and easy to follow.

**Weaknesses:**

PLL itself is a noisy learning case, so it is very confusing to name Noisy PLL as a new noisy learning case. Also, it is worth thinking about whether PLL is learnable even though assuming that the ground-truth resides in the candidate label set, let alone removing the constraint as a new setting NPLL.

The simulated data in the experiments can not well reflect the real-world scenarios, thus It is meaningless to compare the results on these datasets.

BTW, I didn't see any novelty in the proposed framework. It is time to stop and think what is a meaningful research, not just pursuit for paper quantities with meaningless work.

**Questions:**

The techiques in the paper still combines those methods of PLL, what is your technique novelty for NPLL?

The dataset in the experiment is not reflectable on real-world scenarios, and the setting of the noise rate is too subjective. How to improve it? If no real-world data can be used for experiments, how to prove the value of NPLL setting and your proposed method without theorectical guarantee?

---

> ### Author Response · Authors · 2023-11-17
>
> We thank the reviewer for their comments and address their concerns below:
>
> * **Q1: Simulated dataset and not a real world one.**
>     1. There are atleast a dozen papers in top forums (e.g. Neurips, ICML and ICLR) on the same setting in the last couple of years. Hence, we followed and presented the results in the same way, assuming that it is accepted by the community. However, we understand and respect your concern that these datasets are synthetically curated and do not necessarily reflect upon the practical applicability of the PLL/NPLL solutions.
>     1. Your comments encouraged us to explore alternative datasets where the method can be applicable. Last four days, we investigated its applicability in the crowd sourced datasets and performed experiments on three such publicly available real world datasets called Treeversiy, Benthic and Plankton (please refer to the global response). SARI achieves state-of-the-art performance in these real world datasets as well. We demonstrate that these crowd sourced datasets can be approached both from a noise robust learning or a partial label learning perspective. And that adds one practical use case for the PLL/NPLL. We thank you for your comments, which led us to this exploration. We hope, this changes your opinion on our paper. We would be happy to clarify any further queries from your end.
>     1. The quick initial experiments also showcase the adaptability of SARI on varying datasets and scenarios.
>     1. Further, we refer you to a recent work S-CLIP [F], a semi-supervised learning method for training CLIP that utilizes additional unpaired images. This work employs PLL in their pseudo-labelling pipeline, demonstrating another interesting, real-world use case.
>
>
> * **Q2: Novelty of proposed framework.** We request the reviewer to refer to the global response.
>
> [F] Mo, Sangwoo, et al. "S-CLIP: Semi-supervised Vision-Language Learning using Few Specialist Captions." Thirty-seventh Conference on Neural Information Processing Systems. 2023.

---

### Official Review · Reviewer_WW8x · 2023-10-31

**Soundness:** 3 good
**Presentation:** 3 good
**Contribution:** 2 fair
**Rating:** 3
**Confidence:** 4

**Summary:**

This paper proposes a new framework called SARI for noisy partial label learning (NPLL) that combines the benefits of both average-based and identification-based strategies through the utilization of a weighted nearest neighbor method. The proposed framework achieves a good performance compared with the existing state-of-the-art methods, occasionally delivering impressive results, especially in fine-grained classification scenarios. Ablation studies and quantitative experiments convincingly showcase the effectiveness of this approach. Overall, this work is characterized by its simplicity and reader-friendly nature, ensuring accessibility and comprehension for a wider audience.

**Strengths:**

1. This paper introduces a new noisy partial label learning method called SARI, which performs label disambiguation by the weighted nearest neighbor method.
2. This work is straightforward and easily understandable, making it reader-friendly for potential readers.
3. Extensive experimental results demonstrated the effectiveness of the proposed SARI method.

**Weaknesses:**

The paper focus on an interesting noisy partial label learning problem and has several issues that can be improved:

(1) The author scrutinized the deficiencies inherent in current methodologies, highlighting their complexity, warm-up requirements, and error propagation. However, it remains ambiguous how these shortcomings are tackled within the proposed method by the authors.

(2) The proposed method is easy to understand but lacks novelty. It seems no technological innovation in this approach instead of integrating the existing technologies.

(3) The proposed method appears to heavily depend on parameter selection, rendering the approach more empirical than technical.

**Questions:**

(1) My main concern about the paper lies in the novelty: The proposed method seems just integrates existing technologies: like small loss trick, label smoothing, mixup, and consistency regularization by using weak and strong augmentation, which leads to the novelty limited.

(2) The proposed method computes the pseudo-laebl $\hat y_i$ using weighted KNN form the entire dataset, which make the approach memory consuming and time consuming.

(3) In noisy partial label learning, the true label may not in the candidate label set. Thus, the computation of the pseudo-laebl $\hat y_i$ and class posterior probabilities $\hat q_c$ can be noisy, since the assumption proposed in the paper ‘samples in the neighbourhood have the same class label’ can hardly held.

(4) Many experimental results are missing in Table 1, Table 3, and Table 4.

(5) In ablation study, when k=25, the performance improves in $\eta=0.3$ while degrades $\eta=0.5$. However, when k=50, one degrades while the other improves. Can you explain why?

(6) Please see the weakness above.

---

> ### Author Response · Authors · 2023-11-17
>
> We thank the reviewer for their detailed feedback and address their concerns below:
>
> * **Q1: Novelty.** Please refer to the global response. If minimal and existing ideas work well, it would be unjust to push for unnecessary complexities and fancy architectures.
>
> * **Q2: On complexity, warm-up requirements, and error propagation. It remains ambiguous how the shortcomings are tackled within the proposed method :**
>     1. Our approach does not need warm up using a base PLL method. Many methods like ALIM first train PICO for several epochs and then add upon it.
>     1. SARI is a single branch network with a single loss function. It uses standard regularization techniques.
>     1. Reliable image-label pairs and label smoothing makes our method significantly robust to noise. Furthermore, in every iteration we freshly pick the reliable image-label pairs. In contrast, several other methods (like Pico+, IRNet etc.) maintain an explicit state/probability vector, which is sequentially updated at every iteration. Our approach is free of such state maintenance and hence reduces error propagation.
>     1. Label smoothing helps to reduce the impact of noise caused by sample-selection bias, which is another known reason for error accumulation [D] [E].
>
> * **Q3: K-NN is memory and time consuming.**  We would like to emphasize that the K-NN is only used in the training phase. We use FAISS library for optimized nearest neighbour computation. On the CIFAR-100 dataset, our model trains within 3 hours. In contrast, the base PiCO model training takes over 10 hours on the same GPU. Please note that several methods build on top of the PiCO model and may inherit the slow training. We are committed to release the code post paper acceptance to strengthen our case.
>
> * **Q4: Assumption of ‘samples in the neighbourhood have the same class label’.** We assume that the samples of the same class would be closer in feature space. As long as most samples in the neighbourhood contain the ground truth, the pseudo-label will be computed correctly. We perform Partial Label Augmentation which can potentially add the correct label into the partial set. As time progresses, this leads to better pseudo-label computation.
>
> * **Q5: Many experimental results are missing in Table 1, Table 3, and Table 4.** We do a comprehensive set of experiments, compared to some of the other papers. For the matching configuration, we take the numbers from the corresponding original paper. Since the codes were not publicly available at the time of submission, we could not compute the results for other configurations. However, given the available results, there is consistent evidence on the superior performance of SARI.
>
> * **Q6: In ablation study, when K=25, the performance improves in $\eta =0.3$ while degrades $\eta=0.5$. However, when $K=50$, one degrades while the other improves.** The variance remains within 1pp, so results are largely similar. In a highly noisy scenario, the highest value of $K$ seems to benefit the performance, as it would allow to have a larger consensus.
>
>
> [D] Michal Lukasik, Srinadh Bhojanapalli, Aditya Menon, and Sanjiv Kumar. Does label smoothing mitigate label noise? In International Conference on Machine Learning, pp. 6448–6458. PMLR, 2020.
>
> [E] Jian et al. Mentornet: Learning data driven curriculum for very deep neural networks on corrected labels. ICML 2018

---

> > ### Comment · Reviewer_WW8x · 2023-11-23
> >
> > Thanks for your response. I still think integrating existing tricks as the new method is contribution-limited for this conference. Thus, I will keep my score.

---

### Official Review · Reviewer_YQoY · 2023-11-02

**Soundness:** 3 good
**Presentation:** 2 fair
**Contribution:** 3 good
**Rating:** 5
**Confidence:** 3

**Summary:**

The author proposes a new approach in the field of weakly supervised learning in the domain of NPLL. He argues that the current methods three main challenges: 1. complexity 2. the need for warm-up 3.error propagation. In response, the author proposes the SARI model, which integrates the advantages of Average-based strategies and Identification-based strategies. Specifically, the author employs K-nearest neighbors(KNN) to obtain weighted pseudo-labels for unlabeled samples. Then, a classifier is trained using these pseudo-labels, and the training process is enhanced with techniques such as smoothing and consistency regularization to improve robustness. Finally, the author updates the existing labels of the KNN with highes probability class predicted by the current model.

**Strengths:**

1）The author conducts an analysis of the limitations in previous methods for NPLL and identifies complexity, the need for warm-up, and error propagation as the three main challenges in this field. The author's arguments are well-founded and reasonable.

2）The structure of the paper is reading friendly and easy to understand.

3）The author's experiments are comprehensive and thorough.

**Weaknesses:**

1.The author only compares their method on ResNet 18. Given the fact that of several updated backbone networks has been proposed in recent years. It would provide a more comprehensive evaluation of the model's performance under different backbone architectures.

2.I doubt whether the author really addresses the above three issues. The author mentions that the previous method can cause error propagation. However, KNN classifiers according to the overall features from the current encoders, and if there is too much noise features, KNN will also accumulates noise. In addition, the author mentions that the previous method has too many hyper parameters, but according to the author's proposed method, $ K, \delta, \gamma, \lambda$ and other hyper parameters are also required. Further more, according to the ablation study, Mix-up and CR significantly improves the performance. I doubt that the advantage of the author's model rooted by these existing techniques rather than new training methods, which reduces the innovation of this paper.

3.This article needs a more professional writing language.

**Questions:**

As the problems in the weakness.

**Details Of Ethics Concerns:**

I have no ethics concerns.

---

> ### Author Response · Authors · 2023-11-17
>
> We thank the reviewer for their thoughtful feedback and address their concerns below:
>
> * **Q1: Results with other backbones.** In alignment with prior research and for fair comparisons, we limited our experiments to the resnet18 backbone. However, as per the reviewers request we conducted experiments with two other backbones and present the results below. We use the same parameters as in ResNet18 experiments.
>
>     | CIFAR-100| $q=0.05$, $\eta=0.3$ |
>     | --- | ----------- |
>     | Wide-ResNet-34x10 | 81.3 |
>     | DenseNet-121 | 78.75 |
>     | ResNet-18 | 77.61 |
>
>     The performance  improves over the resnet18 backbone in both the cases. We will add experiments with more backbones in the final version of the paper.
>
> * **Q2: Hyperparameters.** SARI has four key hyperparameters $K$, $\delta$, $\lambda$ and $r$ ($\gamma$ is derived from $\delta$).
>     1. We keep the same parameters in all experiments of the paper, across datasets and noise settings.
>     1. Barring extreme values, the performance remains similar with change in variables (variance within 1-2pp, Table 5).
>     1. Even with suboptimal parameters SARI outperforms previous state of the art methods.
>     1. We performed experiments on three new datasets (see global response). With the same parameters we achieve state of the art performance on them as well. Showcasing the simplicity and adaptability of our approach.
>
> * **Q3: On complexity, warm up, error propagation.**
>     1. Our approach does not need warm up using a base PLL method. Many methods like ALIM first train PICO for several epochs and then add upon it.
>     1. SARI is a single branch network with a single loss function. It uses standard regularization techniques.
>     1. Reliable image-label pairs and label smoothing makes our method significantly robust to noise. Furthermore, in every iteration we freshly pick the reliable image-label pairs. In contrast, several other methods (like PiCO+, IRNet etc.) maintain an explicit state/probability vector, which is sequentially updated at every iteration. Our approach is free of such state maintenance and hence reduces error propagation.
>     1. Label smoothing helps reduce the impact of noise caused by sample-selection bias, which is another known reason for error accumulation [D] [E].
>     1. Most previous PLL methods also utilize CR/Mix-up
>
>
> [D] Michal Lukasik, Srinadh Bhojanapalli, Aditya Menon, and Sanjiv Kumar. Does label smoothing mitigate label noise? In International Conference on Machine Learning, pp. 6448–6458. PMLR, 2020.
>
> [E] Jian et al. Mentornet: Learning data driven curriculum for very deep neural networks on corrected labels. ICML 2018

---

> > ### Comment · Reviewer_YQoY · 2023-11-23
> >
> > I do not think that the novelty is sufficient according to the response and I keep my socre 5.

---

### Author Response · Authors · 2023-11-17

### **1. Novelty**

- We would like to point to the reviewers to the developments in the area of domain generalization. Over 150 papers were published in top forums [A], many showing novel architectures and methodologies. Only to find later that none of them really improves over ERM (a simple cross entropy baseline). [B]

- As stated in the paper, the novelty of our paper is to suggest that state-of-the-art performance can be achieved for PLL/NPLL without many of the proposed complexities in prior art, like warm-up, contrastive learning, etc. We also present thorough experiments with notable performance gains.

- In contrast, e.g., PiCO+ uses a two-branch network. One classifier and one momentum encoder. The encoder further branches into a trainable classifier and MLP layers. Contrastive learning is used to train one of the branches. Overall,  it uses five different loss terms to train different components. There are a plethora of loss weighting terms and other hyperparameters. It also employs regularization techniques like mix-up. Some recent methods, like IRNet, add more sophistication as they build upon such base architectures.

- If a simpler alternative works well, should we be adding unnecessary complexities to the method? Would that be meaningful research?

- We urge the reviewers and the area chair to not hold the baseline nature of our algorithms against us. We view it as a strength of our work.


### **2. Results on practical real world datasets (Reviewer EnLY)**

- We show the effectiveness of our approach in real-world annotation settings; we consider three crowdsourced datasets: Treeversity#6, Benthic, Plankton [C].

    1. **Treeversity** is a publicly available dataset of plant images, crowdsourced by the Arnold Arboretum of Harvard University.
    1. **Benthic** is a collection of seafloor images containing flora and fauna. Each image has multiple annotations provided by domain experts.
    1. **Plankton** consists of underwater plankton images annotated by multiple scientists.

- We construct the partial label by considering all annotations given by the human annotators (assuming 10 human annotations are available per image). In the Treeversity dataset, for a few images, all labels were incorrect (simulating the noisy scenario). We use a pre-trained ResNet50 backbone to allow a fair one-to-one comparison with the prior efforts. Please note that prior efforts [C] pose this as a noise-robust learning problem; we tackle it as a partial label learning problem. As the only key change, we utilize the normalized annotation label frequency per image in the weighted KNN step. We present average accuracies using a 3-fold cross-validation below:

    || Treeversity#6 | Benthic | Plankton |
    | --- | ----------- |------|-------|
    | Divide-Mix|0.7784 $\pm$ 0.0052| 0.7172 $\pm$ 0.0121 | 0.9179 $\pm$ 0.0051 |
    | ELR+| 0.7905 $\pm$ 0.0122 |  0.6753 $\pm$ 0.0176|0.9176 $\pm$ 0.0091 |
    | SARI | 0.8156 $\pm$ 0.0028 | 0.7750 $\pm$ 0.0035 |0. 9004 $\pm$ 0.0050 |

- SARI achieves state of the art performance on Treeversity and Benthic datasets. On Benthic, we achieve 6% performance gains. On the Plankton dataset we achieve near SOTA performance.

[A] Wang, J., Lan, C., Liu, C., Ouyang, Y., Zeng, W., Qin, T.: Generalizing to unseen domains: A survey on domain generalization. arXiv:2103.03097 (2021)

[B] Ishaan Gulrajani, David Lopez-Paz. In Search of Lost Domain Generalization.ICLR 2021

[C] Schmarje, Lars, et al. "Is one annotation enough?-A data-centric image classification benchmark for noisy and ambiguous label estimation." Advances in Neural Information Processing Systems 35 (2022): 33215-33232.